# Aerosol Evolution and Influencing Factor Analysis during Haze Periods in the Guanzhong Area of China Based on Multi-Source Data

**Yanling Zhong** [1] , **Jinling Kong** [1,*], **Yizhu Jiang** [2], **Qiutong Zhang** [1], **Hongxia Ma** [1] and **Xixuan Wang** [1]

1   School of Geological Engineering and Geomatics, Chang'an University, 126 Yanta Road, Xi'an 710054, China
2   School of Earth Science and Resources, Chang'an University, 126 Yanta Road, Xi'an 710054, China
*   Correspondence: jlkong@163.com

**Abstract:** Aerosols suspended in the atmosphere negatively affect air quality and public health and promote global climate change. The Guanzhong area in China was selected as the study area. Air quality data from July 2018 to June 2021 were recorded daily, and 19 haze periods were selected for this study. The Hybrid Single-Particle Lagrangian Integrated Trajectory (HYSPLIT) model was used to simulate the air mass transport trajectory during this haze period to classify the formation process. The spatial distribution of the aerosol optical depth (AOD) was obtained by processing Moderate-resolution Imaging Spectroradiometer (MODIS) data using the dark target (DT) method. Three factors were used to analyze the AOD spatial distribution characteristics based on the perceptual hashing algorithm (PHA): GDP, population density, and topography. Correlations between aerosols and the wind direction, wind speed, and precipitation were analyzed using weather station data. The research results showed that the haze period in Guanzhong was mainly due to locally generated haze (94.7%). The spatial distribution factors are GDP, population density, and topography. The statistical results showed that wind direction mainly affected aerosol diffusion in Guanzhong, while wind speed (r = −0.63) and precipitation (r = −0.66) had a significant influence on aerosol accumulation and diffusion.

**Keywords:** typical haze periods; spatial-temporal evolution; AOD; HYSPLIT; PHA; Guanzhong area

## 1. Introduction

Aerosols are a type of colloid formed by solid and liquid particles suspended in the atmosphere that have a diameter between 0.001 and 100 μm in a gaseous medium. Aerosol particles have various physical and chemical properties that affect public health, air quality, and global climate change [1]. Particulate matter that can be inhaled into the human respiratory system will increase atmospheric turbidity and reduce visibility, forming hazy weather [2–4]. Various particles entering the human body are eventually deposited in the bronchi and alveoli, thus causing direct harm to physical health [5,6]. Aerosol prediction can be used to prevent the diffusion of disease, as in the recent COVID pandemic [7]. In addition, changes in radiation caused by various particles can affect the balance of incoming and outgoing energy in the Earth-atmosphere system, leading to changes in dynamic atmospheric processes [8–10]. Therefore, monitoring aerosols is important for both public health and climate change.

Two main methods are available for monitoring aerosols: direct measurements from ground-based observation sites and satellite remote sensing inversions [1,11,12]. Ground-based monitoring data have high accuracy and time continuity but dispersed distributions [13,14]. Therefore, ground-based site data can be used to analyze the temporal variation in aerosols [15,16]. Satellite observations of atmospheric aerosols can reveal the spatial characteristics of a continuous distribution [17–19]. In recent years, several studies have been conducted to analyze aerosol changes and influencing factors.

In 2007, Frank et al. [20] used MISR data to analyze changes in AOD over the Mojave Desert in southern California and suggested that significant differences in AOD in the region can be attributed to adjacent urban sources (Rogers Dry Lake, which is close to the Los Angeles metropolitan area) and local sources (Bristol Dry Lake, which hosts mineral extraction). Du et al. [2] used the Nested Air Quality Prediction Model System (NAQPMS) and HYSPLIT model to analyze the aerosol characteristics in Beijing during six haze periods from November 15 to December 15, 2016. The results show that air mass migration has a significant influence on haze in this region. Although this study upgraded the time scale to a haze period lasting several days, it did not conduct a detailed analysis of its change rules and reasons. Barik et al. [21] used MODIS and MISR Level 3 aerosol products before the monsoon season from 2007 to 2016 combined with rainfall, wind speed, and other data to analyze the relationship between spatial-temporal variations in AOD and meteorological and ground parameters in the Indian subcontinent based on the spatial-temporal Mann–Kendall model and pixel-based multiple linear regression methods. The results suggest that analyzing periodic variations in AOD can provide useful insights on environmental pollution. Li et al. [22] conducted an in-depth study of the distribution and changes in AOD in the Taklimakan Desert and its surrounding areas. The MCD19A2 AOD product, MOD13Q1 normalized difference vegetation index product, and meteorological data were used to study the correlation between the AOD and various influencing factors in the Taklimakan Desert and its surrounding areas based on the spatiotemporal distribution, periodic variation, and influencing factors of AOD. They found that sandstorms affect the seasonal distribution of AOD in the Taklimakan Desert and relative humidity affects the spatial distribution of AOD. However, few studies have focused on the trajectory of air mass migration and the high time-resolution change of aerosols.

The above studies show that fully exploiting appropriate models and meteorological data to explore spatiotemporal changes in aerosols has gradually become a research hotspot in recent years [23–26]. However, problems remain in the current research: (1) research mainly focuses on coastal, desert, and other areas with a single topography, while few studies have focused on areas with complex underlying surfaces, which is problematic because spatiotemporal variations of aerosols vary under the influence of different topographic factors [27,28]. (2) In the above studies, the analysis of aerosol evolution was mostly performed on large time scales, such as monthly, seasonal, and even interannual time scales, which are not suitable for the analysis of certain weather phenomena, such as haze days that only last for a few days. Therefore, this study selected 19 typical haze periods by counting the daily air quality data of five cities with complex underlying surfaces in the Guanzhong area of China from July 2018 to June 2021. The DT method was used to invert the AOD of the daily scale 550 nm land background during the haze period using MODIS images with a short revisit period. The 550 nm wavelength is associated with major radiation effects because it is near the peak of the solar spectrum. HYSPLIT was used to simulate the moving trajectory of the air mass in the study area, and the context and evolution rules of aerosol particles during the haze period were confirmed. This study also analyzed the spatial distribution of aerosols and their influencing factors by combining GDP, population density, and digital elevation model (DEM) data and evaluated the drivers of air quality index (AQI) changes over time using precipitation, wind speed, and wind direction data from NOAA-NCEI.

## 2. Materials and Methods

### 2.1. Study Area

The Guanzhong area is located in the central part of Shaanxi Province, China, and it includes the cities Xi'an, Xianyang, Weinan, Baoji, and Tongchuan. The Guanzhong area has a temperate monsoon climate, which is dominated by northwest winds in winter, when haze occurs frequently. The geographical location of the study area is shown in Figure 1.

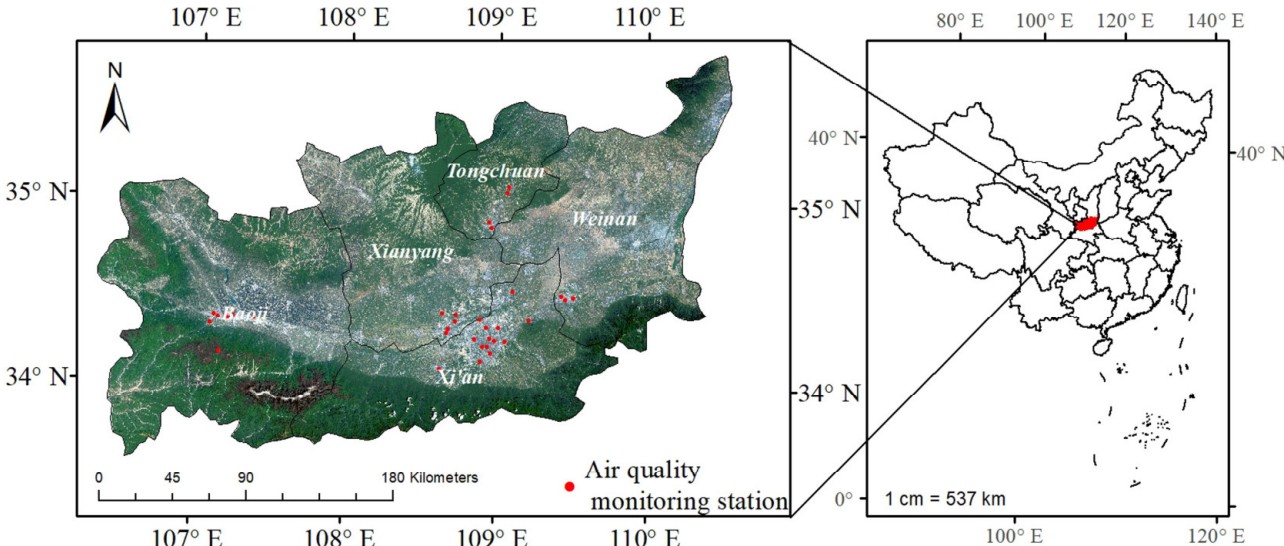

**Figure 1.** Geographic location map of Guanzhong, synthesized from Landsat 8 true color images. The red dots are the locations of the air quality monitoring stations.

The Guanzhong area is the economic center of Shaanxi Province and even Northwest China, and it presents a high level of human activities and developed heavy industries. Therefore, the atmospheric environment is greatly affected by human activities and the heavy industry-based economy. Under such severe air pollution conditions, it is of great significance to study the temporal and spatial evolution of aerosols and their influencing factors in Guanzhong.

### 2.2. Research Process

The AQI was used as a reference to determine the duration of the haze period. The HYSPLIT model was used to analyze the forward and backward trajectories of the air mass during the haze period to determine the source and destination of aerosol particles. MODIS data were used to invert the AOD of the study area during the haze period to analyze the spatial distribution characteristics and changes in aerosols. DEM, GDP, and population density data were used to analyze the factors influencing the spatial distribution of aerosols during the haze period. The time period when the AQI significantly increased or decreased was selected from the duration of the haze period, and the wind speed, wind direction, and precipitation data in this time period were combined to analyze the influencing factors of AOD at the same location over time. The process is shown in Figure 2.

### 2.3. Data

#### 2.3.1. Remote Sensing Data

This study used MODIS02 L1B 1 km remote sensing data, which were downloaded from NASA's official website (https://ladsweb.modaps.eosdis.nasa.gov/, accessed on 8 October 2021). These data have a temporal resolution of 1 d, a spatial resolution of 1 km, and 36 bands; thus, they are suitable for AOD inversion [29–32]. Therefore, this study used MODIS products to retrieve AOD data for the study area.

#### 2.3.2. Global Data Assimilation System (GDAS) Data

GDAS data were used to support the HYSPLIT model [33,34]. GDAS data include temperature, air pressure, relative humidity, and horizontal and vertical wind speeds. The forward versus backward trajectories of the HYSPLIT model mainly used horizontal and vertical wind speed data in the GDAS. The GDAS interpolates global $1° \times 1°$ data into a conformal map projection. Approximately 571 Mb of data have been collected every 7 days from January 2005 until now, and these data are updated monthly.

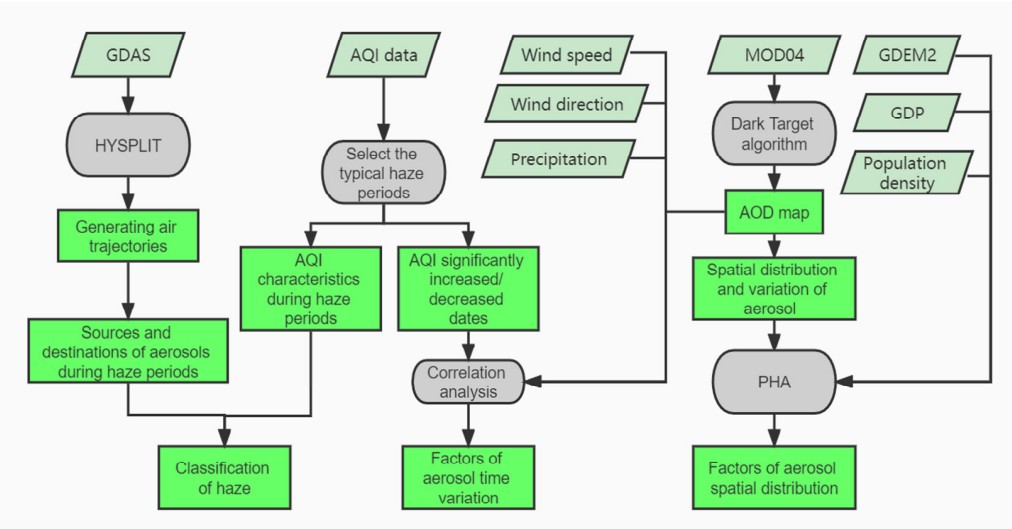

**Figure 2.** Flow chart of research on the aerosol evolution law and influencing factors during the typical haze period in Guanzhong.

### 2.3.3. Topography Data

The topographic data were obtained from the Global Digital Elevation Model version 2 (GDEM 2). The data were jointly released by the Ministry of Economy, Trade and Industry (METI) of Japan and NASA.

The satellite-borne thermal emission and reflection radiometer on NASA's Terra satellite generates images from stereo image pairs. These stereo pairs have been used to generate a single-scene (60 km) DEM since 2001, with root mean square errors typically between 10 m and 25 m [35,36]. The topography of the study area is shown in Figure 3:

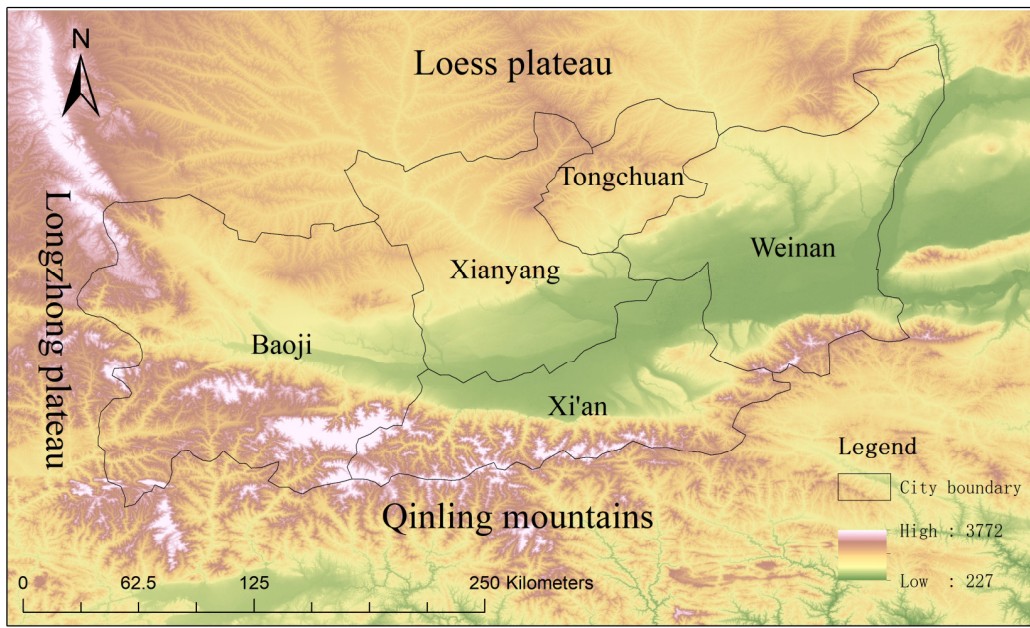

**Figure 3.** DEM map of Guanzhong and the surrounding area. The area enclosed by the black curve is Guanzhong, which is north of the Qinling Mountains, east of the Longzhong Plateau, and south of the Loess Plateau. The area is high on three sides, low on one side, and flat only to the east. Because its terrain is high on three sides and low in the middle, air does not easily diffuse, resulting in the accumulation of aerosol particles and an effect on air quality.

### 2.3.4. Air Quality Data

The air quality data were obtained from the national control point of the air quality monitoring stations of the Ministry of Ecology and Environment, PRC, and the distribution of the monitoring stations is shown in Figure 1. The AQI of Guanzhong from July 2018 to June 2021 was counted daily, and the average value of each city's daily air quality monitoring stations was taken as the air quality value of the city on that day. A statistical graph of these data was drawn as shown in Figure 4:

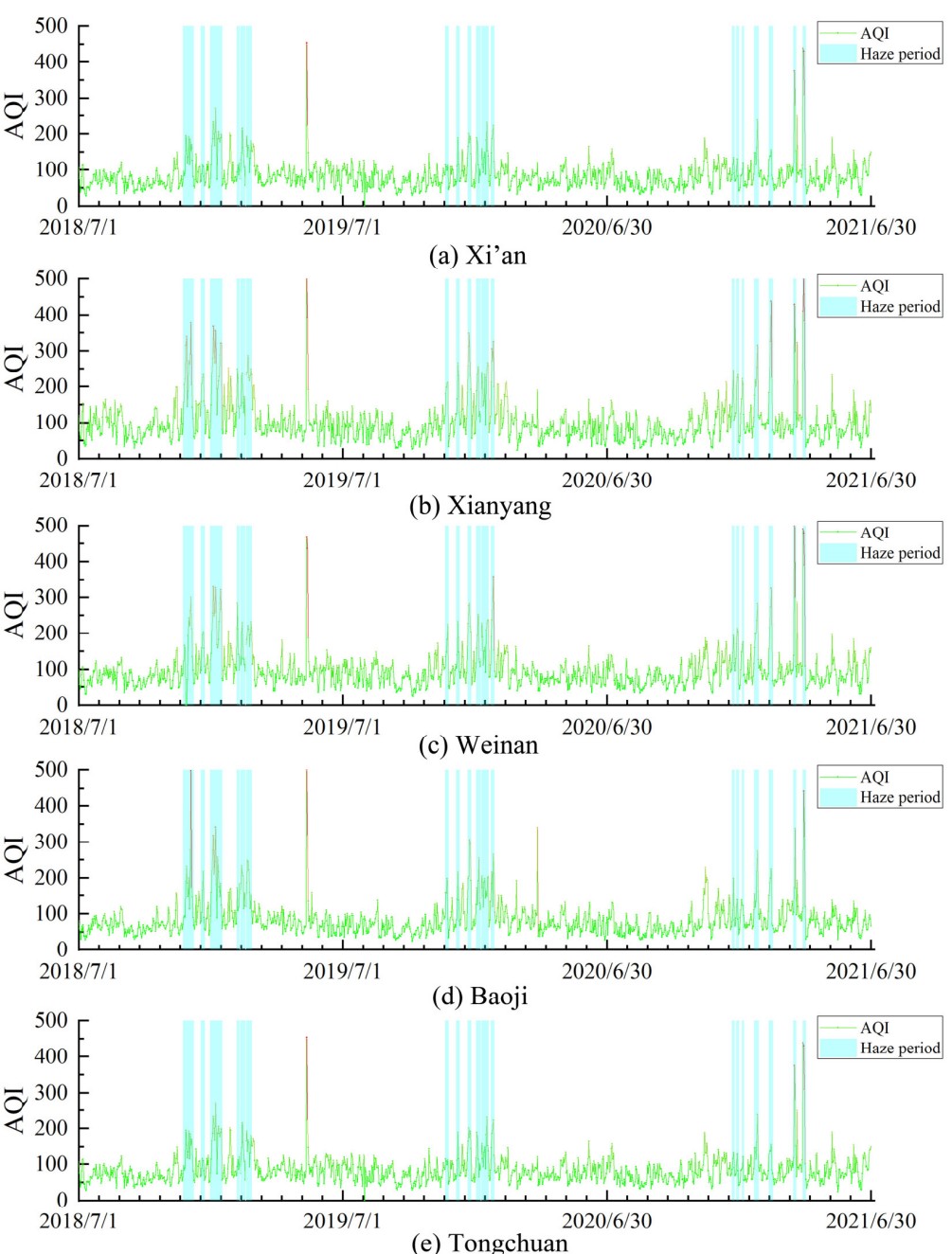

**Figure 4.** Historical AQI of 5 Guanzhong cities. (**a**) is Xi'an, (**b**) is Xianyang, (**c**) is Weinan, (**d**) is Baoji, (**e**) is Tongchuan.

Statistical analyses have shown that the air quality in each haze process has a changing trend from low to high to low. In this study, the period when the AQI of more than half of the cities (3 or more) in the Guanzhong area reached more than 200 was selected as the haze

period. Considering the situation of the 5 cities, 19 haze period processes were determined and are marked in Figure 4.

### 2.3.5. Other Data

The *Statistical Yearbook of Shaanxi Province (2019–2021)* issued by the Shaanxi Provincial Bureau of Statistics was used to calculate the GDP and population density data for the study areas. Wind speed, wind direction, and precipitation data were also used to analyze factors that significantly increased and decreased the AOD. Data were obtained from the National Oceanic and Atmospheric Administration (NOAA) National Centers for Environmental Information (NCEI) website (https://www.ncei.noaa.gov/, accessed on 9 October 2021).

### *2.4. Methods*
### 2.4.1. DT Method

MODIS daytime AOD products include MOD04 and MCD19. The AOD inversion of MOD04 uses the DT algorithm. MOD04 has two products with 3 km resolution and 10 km resolution, which is low resolution. However, the multi-angle implementation of atmospheric correction (MAIAC) algorithm used by the MCD19 product [37,38] was not robust in the study area. For these reasons, MODIS02 L1B 1 km data were finally selected and robust DT-retrieved AOD values were used for the aerosol analysis.

Principle of the DT method [39,40]: In areas covered by high vegetation, wet soil, and water bodies, the surface reflectance of the visible red light (0.66 μm), blue light (0.47 μm), and mid-infrared (2.1 μm) bands are related as follows:

$$\begin{cases} 0.0 < \rho^*_{2.1} < 0.4 \\ \rho^*_{2.1} \approx 2\rho_{0.66} \\ \rho^*_{2.1} \approx 4\rho_{0.47} \end{cases} \tag{1}$$

DT is used to remove the surface contribution from the apparent reflectance to realize the decoupling of the surface and atmosphere. The AOD value can then be obtained by setting certain aerosol types and atmospheric model parameters and analyzing the atmospheric path radiation term. Then, the AOD is derived from the 6S model [41]. The values of MODIS band1 (red light), band3 (blue light), and band7 (mid-infrared) were used as the spectral parameters. The AOD retrieved using DT is a dimensionless value between 0 and 2, where values closer to 0 indicate a more transparent atmosphere and values closer to 2 indicate an opaquer atmosphere.

As an AERONET (Aerosol Robotic Network) monitoring station is absent in the research area, MODIS04 3 km product is used to verify the aerosol inversion result. Two images at the same time on 25 February 2020 were selected for comparison, and their R = 0.98 was very consistent. It shows that DT is suitable for AOD inversion in the Guanzhong area.

### 2.4.2. HYSPLIT

The capabilities of HYSPLIT include calculating air mass trajectories and simulating complex diffusion and deposition [42,43]. It was originally developed based on a collaboration between NOAA and the Australian Bureau of Meteorology. HYSPLIT assumes that the trajectory of the particle moves with the wind field and is the integral of the particle in space and time. The vector velocity of the particle position is obtained via linear interpolation in time and space. The basic concept is as follows:

$$\begin{cases} P'(t + \Delta t) = P(t) + v(P,t) \cdot \Delta t \\ P(t + \Delta t) = P(t) + 0.5[v(P,t) + v(P',t + \Delta t)] \cdot \Delta t \end{cases} \tag{2}$$

where $P'(t + \Delta t)$ is the first guess point, which is obtained by position $P(t)$ and adding the product of velocity $v(P,t)$ and time $\Delta t$. Position $P(t + \Delta t)$ at the next time is obtained by position $P(t)$ and adding the product of the mean velocity (from velocity $v(P,t)$, velocity

v(P',t + Δt)) and time Δt. The HYSPLIT model in this study was run with the support of GDAS data. When the HYSPLIT model is used to analyze air mass trajectory, the trajectory is mainly divided into forward and backward trajectories. The forward trajectory is used to analyze the trajectory of the air mass moving forward from its current position, while the backward trajectory is used to analyze where the air mass originated from.

### 2.4.3. Perceptual Hashing Algorithm (PHA)

The role of the PHA is to generate a "fingerprint" string for each image and then compare the fingerprints of different images [44,45]. Closer results correspond to more similar images. The PHA is widely used in sound and image detection. When analyzing the aerosol spatial distribution factors in this study, influencing factors with a more similar distribution to the aerosol distribution indicate a stronger correlation between the factor and aerosol distribution.

The PHA process is as follows.

I.   Resample the image to an $8 \times 8$ size, with 64 pixels in total. The purpose of this step is to eliminate differences in image size, scale, and resolution.
II.  Convert the reduced image to 64 grayscale images. The purpose of this step is to eliminate any differences between the images caused by the use of different color bands.
III. Perform discrete cosine transform (DCT). Due to the strong "energy concentration" property of DCT, the energy of most natural signals (including sound, images, etc.) is concentrated in the low frequency part after DCT. After DCT transformation, the image becomes a $32 \times 32$ matrix.
IV.  DCT reduction. We only need to retain the $8 \times 8$ matrix in the top-left corner, which represents the lowest frequency in the image.
V.   The average of all 64 values is calculated. The $8 \times 8$ matrix and average are compared, and the 64-bit hash value is set to "1" if it is greater than or equal to the average and to "0" if it is less than the average.
VI.  The comparison results are set to a 64-bit string, which is the fingerprint of the image. Comparing fingerprints between images, a smaller number of different characters (Hamming distance) indicates a higher similarity.

## 3. Results and Analysis

### 3.1. Statistical Analysis of Air Quality Data

The air quality changes in the study area were counted, and the course of the 19 haze periods is shown in Table 1.

**Table 1.** Basic situation of 19 typical haze periods.

| Start Year | Duration | Days | Maximum AQI | Average AQI | Times (Average Days) |
|---|---|---|---|---|---|
| | 11.22–12.7 | 16 | 497 (Baoji on 12.3) | 172 | |
| 2018 | 12.16–12.22 | 7 | 234 (Xi'an on 12.20) | 133 | 3 (13.7 days) |
| | 12.29–1.15 | 18 | 367 (Xianyang on 1.3) | 202 | |
| | 2.4–2.8 | 5 | 284 (Xi'an on 2.5) | 143 | |
| | 2.9–2.16 | 8 | 236 (Xianyang on 2.12) | 151 | |
| 2019 | 2.17–2.25 | 9 | 286 (Xianyang on 2.20) | 188 | 6 (6.7 days) |
| | 11.19–11.24 | 6 | 231 (Xi'an on 11.23) | 133 | |
| | 12.4–12.9 | 6 | 266 (Xianyang on 12.7) | 135 | |
| | 12.20–12.25 | 6 | 348 (Xianyang on 12.22) | 223 | |
| | 1.1–1.7 | 7 | 255 (Xianyang on 1.4, Baoji on 1.5) | 151 | |
| | 1.8–1.18 | 11 | 266 (Xianyang on 1.17) | 177 | |
| 2020 | 1.21–1.26 | 6 | 358 (Weinan on 1.25) | 207 | 5 (6.8 days) |
| | 12.19–12.23 | 5 | 244 (Xianyang on 12.22) | 126 | |
| | 12.25–12.29 | 5 | 232 (Xianyang on 12.28) | 128 | |
| | 1.2–1.5 | 4 | 222 (Xianyang on 1.3) | 141 | |
| | 1.19–1.25 | 7 | 315 (Xianyang on 1.24) | 177 | |
| 2021 | 2.8–2.14 | 7 | 438 (Xianyang on 2.12) | 173 | 5 (5.6 days) |
| | 3.14–3.18 | 5 | 500 (Weinan on 3.16) | 199 | |
| | 3.27–3.31 | 5 | 500 (Xi'an and Xianyang on 3.29) | 280 | |

Data for 19 haze periods in the study area were analyzed, and the results showed that the haze process in the study area lasted an average of 7.5 days, with the longest at 18 days and the shortest at 4 days. The statistical results show that Xianyang had the highest number of high AQI values in the haze period at 13 (including a tie), followed by Xi'an (4), Weinan and Baoji (2), and Tongchuan (0). According to the AQI data monitored by the site, the air quality of Xianyang and Xi'an in the study area was the worst, followed by Weinan and Baoji, whereas Tongchuan had the best air quality.

An analysis of the time that the high AQI values occurred in the haze period showed that the highest AQI values in most haze periods are in the middle and late haze periods, as shown in Figure 5a. Only a few of the highest AQI values in the haze period occurred on the first or second day of the haze period, as shown in Figure 5b.

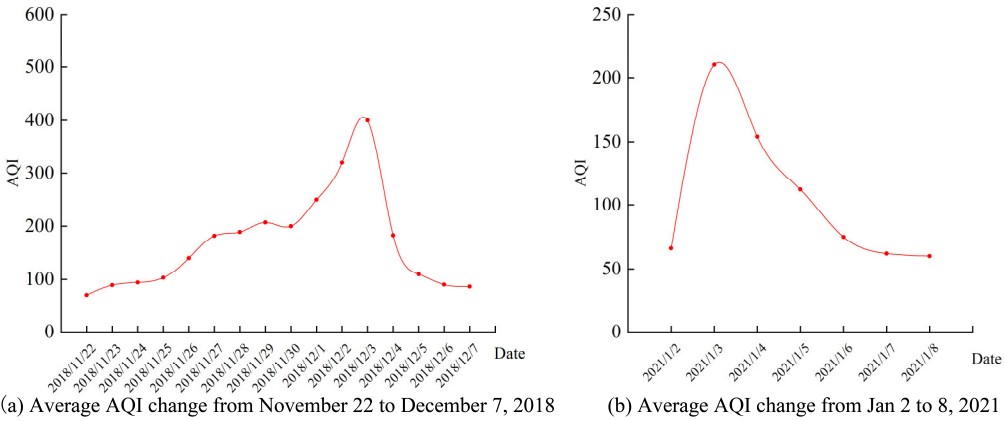

(a) Average AQI change from November 22 to December 7, 2018     (b) Average AQI change from Jan 2 to 8, 2021

**Figure 5.** Changes in AQI during the haze period. It is the average of all sites. (**a**) The maximum AQI occurs in the late period. (**b**) The maximum AQI occurs in the early period.

### 3.2. Classification of Haze Causes

The HYSPLIT model was used to analyze the air mass trajectories in the study area when the AQI increased or decreased significantly, which is convenient for determining where dirty air masses in the haze period come from and where they go. By summarizing the air mass trajectory and AQI variation characteristics during the haze period, the causes of haze are divided into two types: locally generated and externally transported.

The haze period from 22 November to 7 December 2018, was selected as an example to analyze the trajectory characteristics of locally generated haze. According to the historical AQI values of each city, the AQI value increased significantly from 28 November to 29 and decreased significantly from December 4 to 5. The backward trajectory of the air mass on 29 November 2018 and the forward trajectory of the air mass on 6 December 2018 generated by the HYSPLIT model are shown in Figure 6.

As shown in Figure 6a, the trajectory lines of the air mass at heights of 10 and 100 m were all very short, and only part of the trajectory lines of the air mass at a height of 500 m were slightly longer. Therefore, the air mass in Guanzhong moved slowly from November 28 to 29. This phenomenon is consistent with the significant increase in AQI values in Guanzhong during this period, thus indicating that the air mass did not transport aerosol particles generated in the study area over time. Therefore, the wind speed was not sufficient to move internal pollutants in time. Therefore, the haze was considered locally generated.

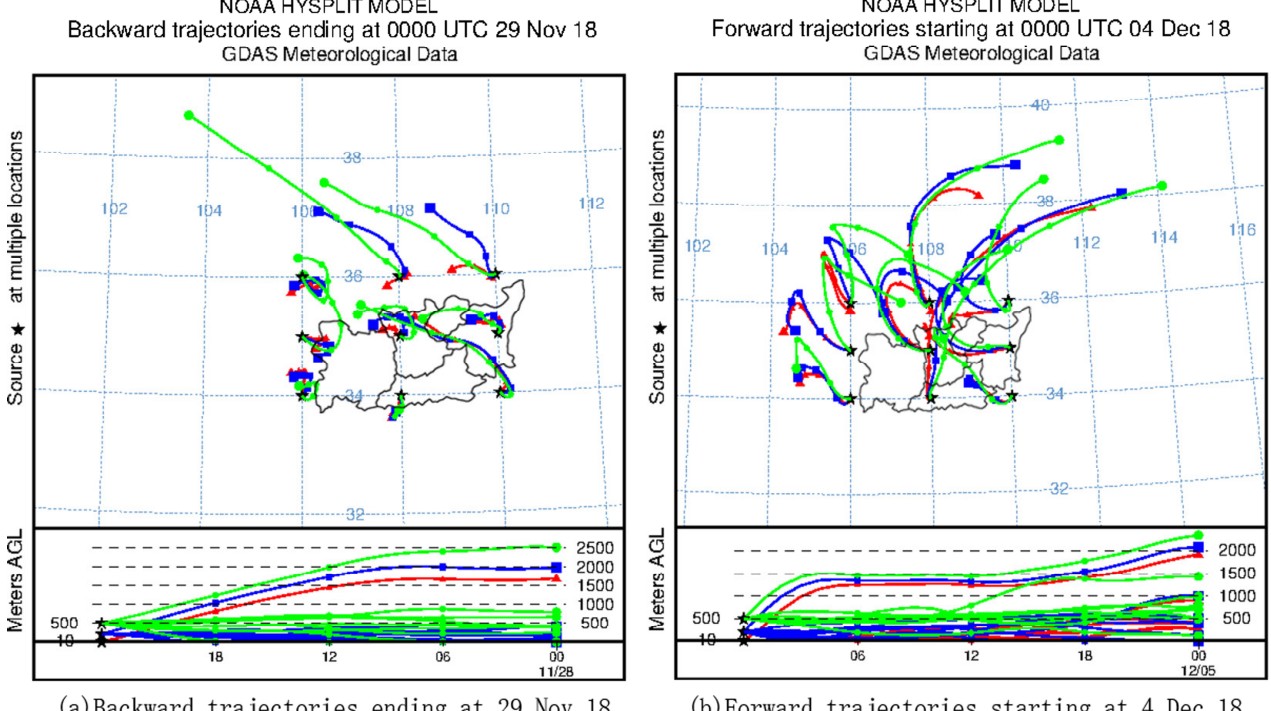

**Figure 6.** Trajectory characteristics of air masses during the locally generated haze period. The red, blue, and green line segments represent the trajectory of the air mass at a height of 10, 100, and 500 m from the surface, respectively. (**a**) shows the track of the air mass in Guanzhong from 28 November to 29 November 2018, and (**b**) shows the track of the air mass in Guanzhong from 4 December to 5 December 2018. Over the same time interval (24 h), a longer trajectory indicates a faster wind speed and vice versa.

The AQI in Guanzhong decreased significantly from December 4 to 5. Figure 6b shows that the western air mass first moved northwest and then reversed to the southeast from the 4th to the 5th, which indicated convective weather. An analysis of historical weather indicated that rainfall occurred from December 4 to 5, which reduced the concentration of aerosol particles. The eastern air mass mainly migrated from east to north, and the DEM data showed that the terrain in this direction is flat; therefore, the migration of the air mass was able to carry away a large amount of aerosol particles, which significantly reduced the AQI value. Finally, the haze period ended under the joint action of these meteorological factors.

The haze period from 2 to 5 January 2021 was selected as an example to analyze the trajectory characteristics of externally transported haze. Figure 7a shows the trajectory of the study area after 2 to 5 January 2021. The trajectory of the air mass in the figure shows that the air mass originated from areas outside Guanzhong, such as the Loess plateau. Historical AQI data show that the AQI suddenly rose to over 200 from 2 to 3 January 2021. The analysis in Table 1 shows that the highest AQI value of this haze period occurred on the second day. During this period, the air mass in the Guanzhong area was mainly from outside. Therefore, the haze period was considered to be of the external transport type.

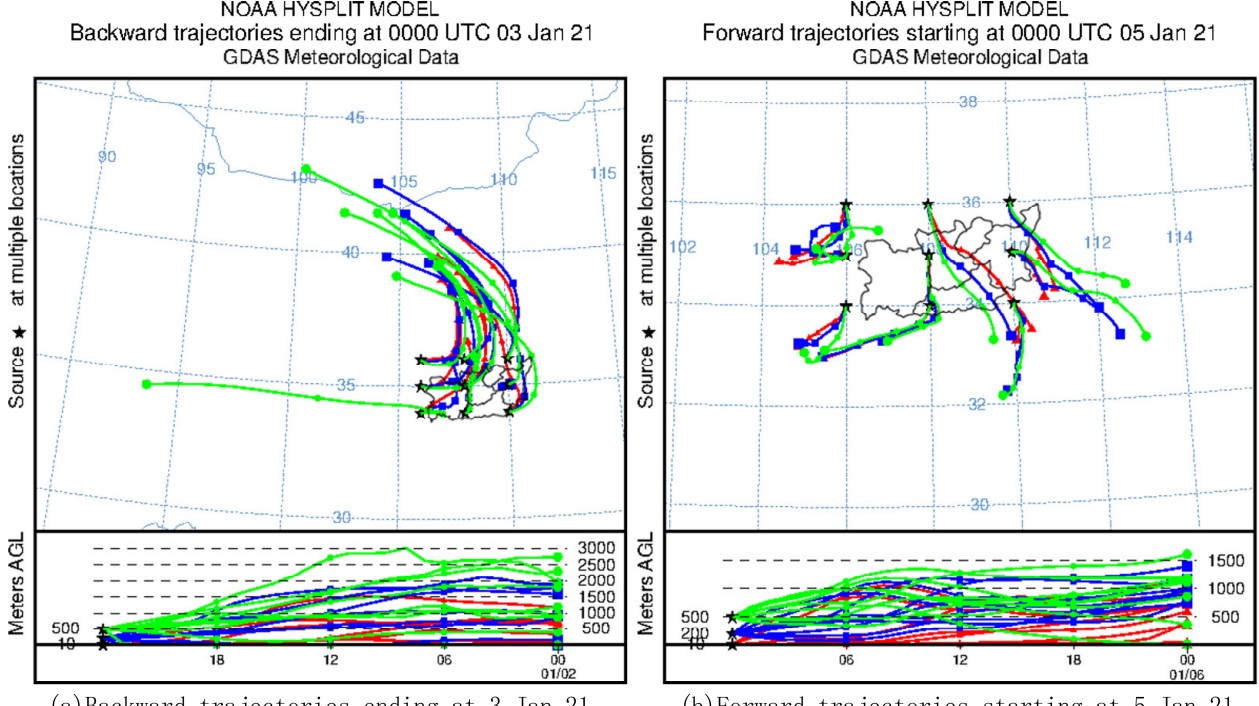

(a) Backward trajectories ending at 3 Jan 21  (b) Forward trajectories starting at 5 Jan 21

**Figure 7.** Trajectory characteristics of the air mass in the externally transported type haze period. The meaning of the colored line segments is the same as those in Figure 6. (**a**) shows the track of the air mass in Guanzhong from 2 to 3 January 2021 and (**b**) shows the track of the air mass in Guanzhong from 5 to 6 January 2021.

The same method was used to analyze the data for 19 haze periods over a total of three years from July 2018 to June 2021. The haze period in the study area was divided into locally generated and external transport types (Table 2).

**Table 2.** Types of haze formation in the study area.

| Type | Air Mass Trajectory Characteristics | Variation Characteristics of AQI |
| --- | --- | --- |
| locally generated | Trajectory of the air mass is very short and migration ability is limited, as shown in Figure 6a. | Maximum AQI is generally produced in the middle or end of the haze. Then, AQI rapidly decreases to a low level, as shown in Figure 5a. |
| External transport | Air masses are transported from distant areas over long distances, as shown in Figure 7a. | The AQI abruptly reaches the maximum value in the early stage of haze and then slowly decreases day by day, as shown in Figure 5b. |

Nineteen haze periods were classified, which showed that 18 haze periods in Guanzhong were locally generated, thus accounting for up to 94.7%. This is closely related to the fact that Guanzhong is the economic center of Northwest China, with a large population density and developed industry.

### 3.3. AOD Spatial Distribution Characteristics and Influencing Factor Analysis

The spatial distribution characteristics of aerosols in the study area were obtained by retrieving AOD data from remote sensing imagery. When the DT is used for inversion, it is affected by clouds and leads to null values. When the cloud cover of a remote sensing image is large, it will affect the authenticity of the rest of the image. Thus, images with

less than 30% cloud cover were selected to analyze spatial distribution characteristics. To ensure the integrity of the images, kriging interpolation was performed on the null value part of the retained images.

The spatial characteristics of the aerosol distribution in Guanzhong were obtained by the seasonal synthesis of all filtered AOD images from July 2018 to June 2021, as shown in Figure 8.

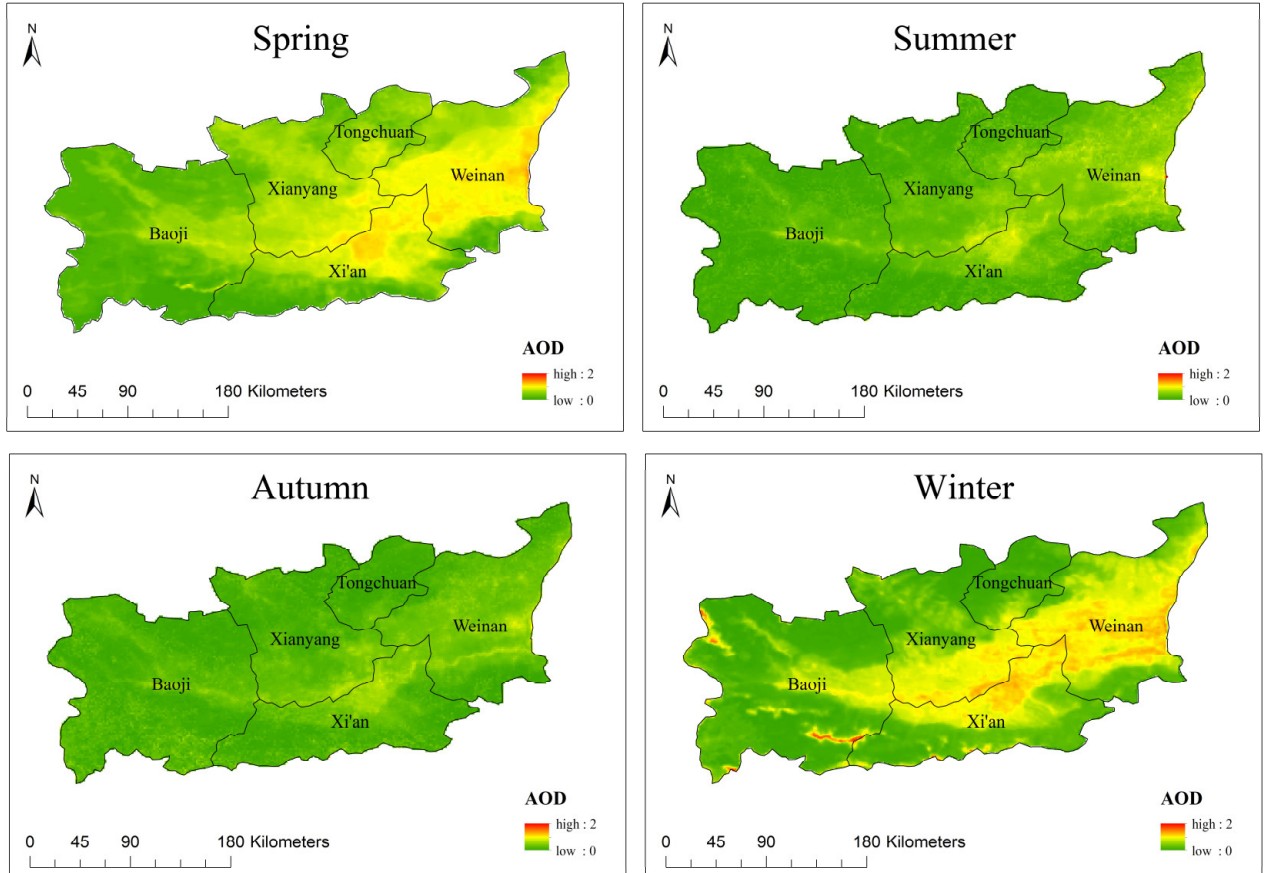

**Figure 8.** Seasonal distribution of AOD. The seasons are divided into spring (March to May), summer (June to August), autumn (September to November), and winter (December to next February).

Figure 8 shows that the AOD in Guanzhong was higher in winter and spring and lower in summer and autumn, thus showing distinct seasonal changes. The average AOD values and standard deviations for the four seasons were winter (avg = 0.39, σ = 0.28), spring (avg = 0.37, σ = 0.22), summer (avg = 0.20, σ = 0.12), and autumn (avg = 0.14, σ = 0.09). The *Statistical Yearbook of Shaanxi Province (2019–2021)*, released by the Shaanxi Provincial Bureau of Statistics, was used to generate statistical data on the GDP and population density for the study area from 2019 to 2021, and the statistical thematic map was produced using the three-year average value, as shown in Figure 9:

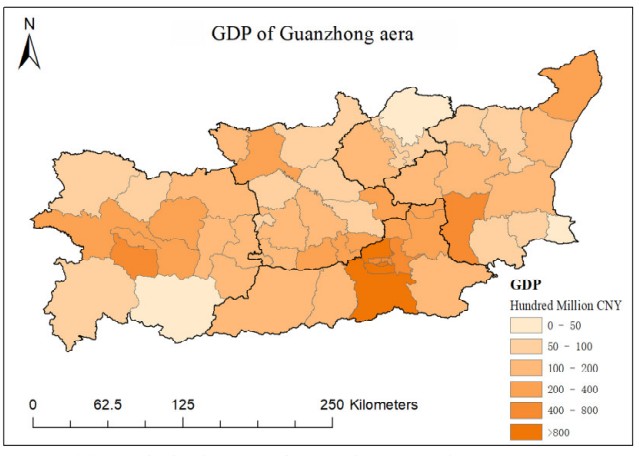

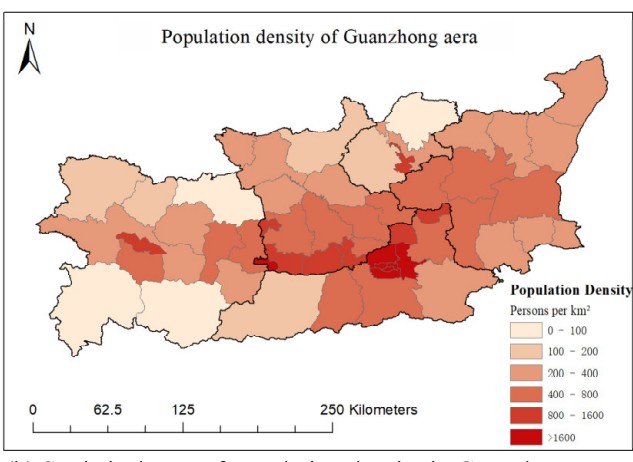

(a) Statistical map of GDP in Guanzhong area　　　(b) Statistical map of population density in Guanzhong area

**Figure 9.** Statistical chart of socioeconomic factors in the Guanzhong area from 2019 to 2021. (**a**) shows the GDP in Guanzhong area, and (**b**) shows the population density in Guanzhong area. The figure shows that the economic and population centers of the Guanzhong area are mainly located in the central and eastern parts.

After synthesizing AOD images during the haze period, PHA detection was carried out using statistical maps of GDP and population density (Figure 9a,b) and DEM (Figure 3), and then the Hamming distance of the fingerprints was calculated. Table 3 presents the results.

**Table 3.** Hamming distance between the AOD and factors influencing the spatial distribution.

| Influencing Factors | GDP | Population density | Topography |
|---|---|---|---|
| Hamming distance | 8 | 13 | 18 |

Table 3 shows that the correlation between the AOD of Guanzhong and the above three indicators from strong to weak is GDP > population density > topography. An analysis of the three factors reveals that the AOD distribution is most similar to the GDP distribution. The spatial distribution of AOD was also similar to the population density, the corresponding AOD values are higher in areas with a larger population density, and vice versa. Guanzhong has high terrain on three sides and low terrain in the middle, which is not conducive to aerosol diffusion. AOD distribution is also correlated with topography.

*3.4. Characteristics of AOD Changes over Time and Analysis of Natural Factors*

An analysis of the historical air quality data of the study area showed that among the 19 haze periods, 27 significant AQI increases and 25 significant AQI decreases occurred. Thus, the influence of wind speed, wind direction, and precipitation on aerosols was analyzed when the AQI increased or decreased significantly.

3.4.1. Wind Speed

The daily average wind speed of each city during the 27 and 25 significant increases and decreases, respectively, in the AQI was determined, and the null values and outliers were eliminated, resulting in a total of 134 valid station data. Wind speed data were obtained from NOAA-NCEI daily meteorological data. A linear function model was established to fit the relationship between the daily average wind speed and AOD change.

$$\Delta AOD \ = \ a + bv \qquad (3)$$

where $\Delta AOD$ represents the change value of AOD in unit time (days), with positive and negative values indicating increases and decreases, respectively; a is an environmental constant that represents the accumulated value of basic pollutants in the natural and social

environments of the study area, although it is affected by rainfall, temperature, industrial output value, human activities, and other factors; and bv is the diffusion capacity of air pollutants under the transport of wind, with b representing the transport coefficient and v representing the wind speed.

The data were substituted to perform the data fitting, and the conclusions are shown in Figure 10:

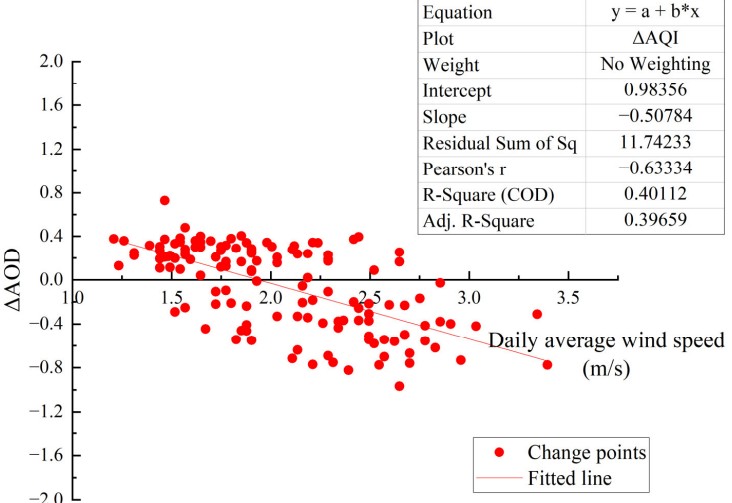

**Figure 10.** Fitting diagram of wind speed and AOD. In the figure, the horizontal axis is the daily average wind speed, the vertical axis is the daily variation of AOD, and the red line is the fitting line.

The constant a obtained by fitting is 0.98356, the migration coefficient b obtained by fitting is −0.50784, and the goodness of fit $R^2$ of the fitted function model is 0.40112. An analysis of the data showed that the Pearson correlation coefficient r between wind speed and ΔAOD is −0.63, thus indicating a negative correlation. A monitoring station with good data integrity was selected as an example to illustrate the change relationship between wind speed and ΔAOD in the haze period, as shown in Figure 11.

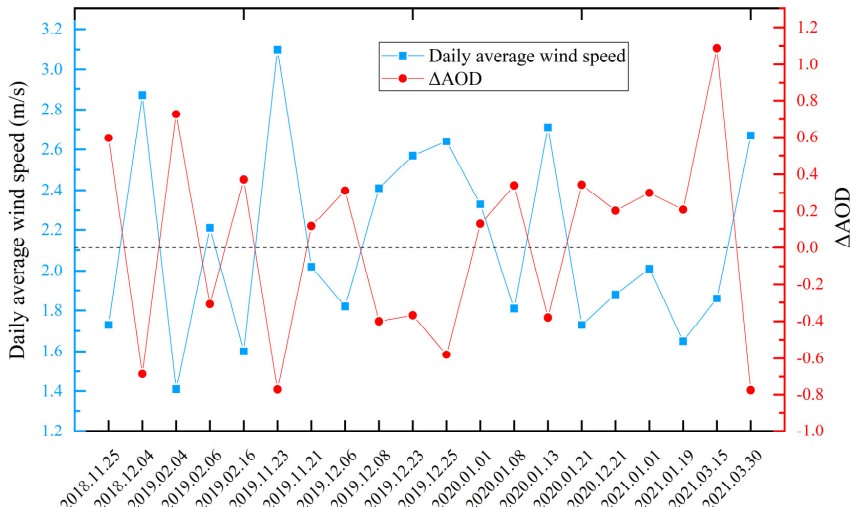

**Figure 11.** Relationship between daily average wind speed v and ΔAOD. The horizontal axis shows the dates, including dates that satisfy both the wind speed data of the monitoring station and the AOD value from the inverted MODIS image, and the significant AQI change date was identified. The left vertical axis is the daily average wind speed, and the right vertical axis is the daily variation of AOD. The black dashed is ΔAOD = 0, which intersects the left axis at 2.1 m/s position.

Figure 11 shows that the ΔAOD and daily average wind speed demonstrate obvious negative correlations. The daily average wind speed of 2.1 m/s is a critical value. When v > 2.1 m/s, the wind speed is conducive to aerosol diffusion, the AOD decreases, and the air quality becomes better. Conversely, when v < 2.1 m/s, the wind speed is conducive to aerosol aggregation, the AOD increases, and the air quality becomes worse.

### 3.4.2. Wind Direction

The wind direction when the AQI increased significantly during the haze period was calculated, as shown in Figure 12a. This is consistent with the climatic characteristics dominated by northwest wind in winter and spring. Because the wind speed is relatively small, the northwest wind makes the aerosol in Guanzhong region gather in the central and eastern part, thus the AOD of Xi'an, Xianyang, and Weinan is higher.

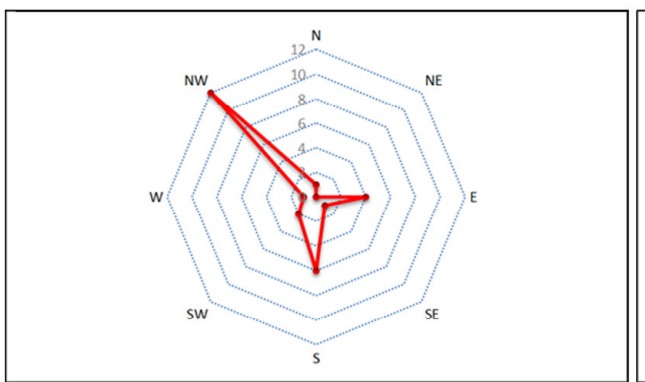
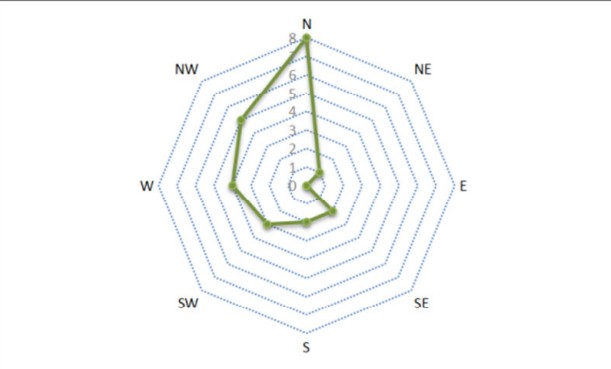

(a) Wind direction when AQI rising    (b) Wind direction when AQI falling

**Figure 12.** Statistical diagram of the wind direction when AQI obviously increases and decreases during the haze period. (**a**) is the wind direction when the AQI rising, and (**b**) is the wind direction when the AQI falling.

An analysis of the factors responsible for the significant decrease in AQI during the haze period shows that the frequency of northwest, west, and southwest winds (12 times) was significantly greater than that of northeast, east, and southeast winds (3 times) (Figure 12b). Plateaus are observed in the north, west, and south of the study area, and westerly winds had a positive effect on reducing the concentration of aerosol particles. Therefore, wind direction also plays a significant role in the reduction of aerosol concentrations in the study area.

### 3.4.3. Precipitation

Because of the need to satisfy the three conditions of obvious AQI change, available precipitation data at the monitoring station, and simultaneous AOD inversion values, only 25 valid data were finally acquired. The fitting results for the precipitation data and AOD change values are shown in Figure 13.

The goodness-of-fit ($R^2$) for the fit was 0.43739. An analysis of the data shows that under the background of precipitation, the Pearson correlation coefficient r between precipitation and ΔAOD was −0.66, thus indicating a negative correlation. The relationship between precipitation and ΔAOD during the haze period was plotted with the data, as shown in Figure 14.

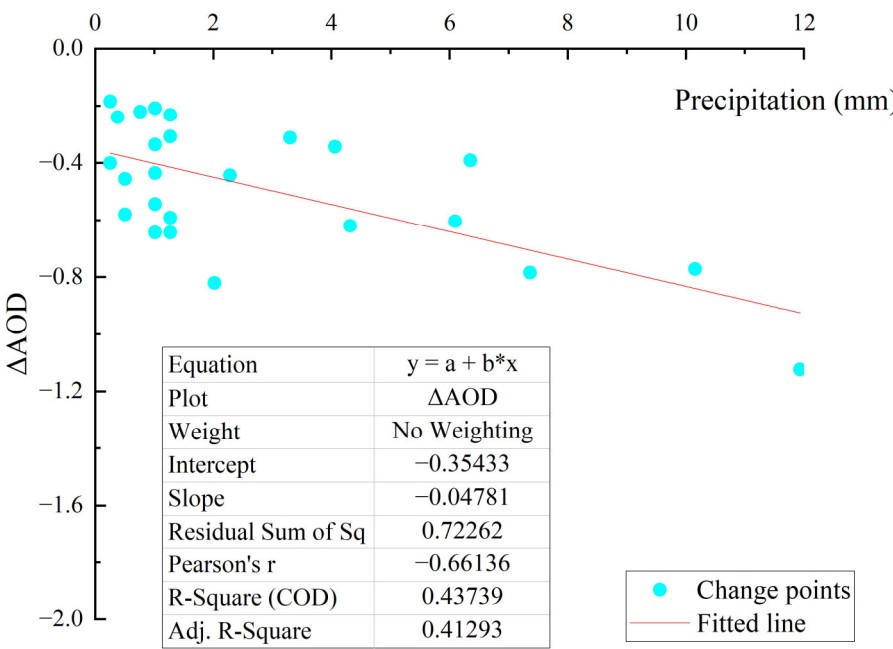

**Figure 13.** Precipitation data and ΔAOD fitting plot. In the figure, the horizontal axis is precipitation, the vertical axis is daily ΔAOD, and the red line is the fitted line.

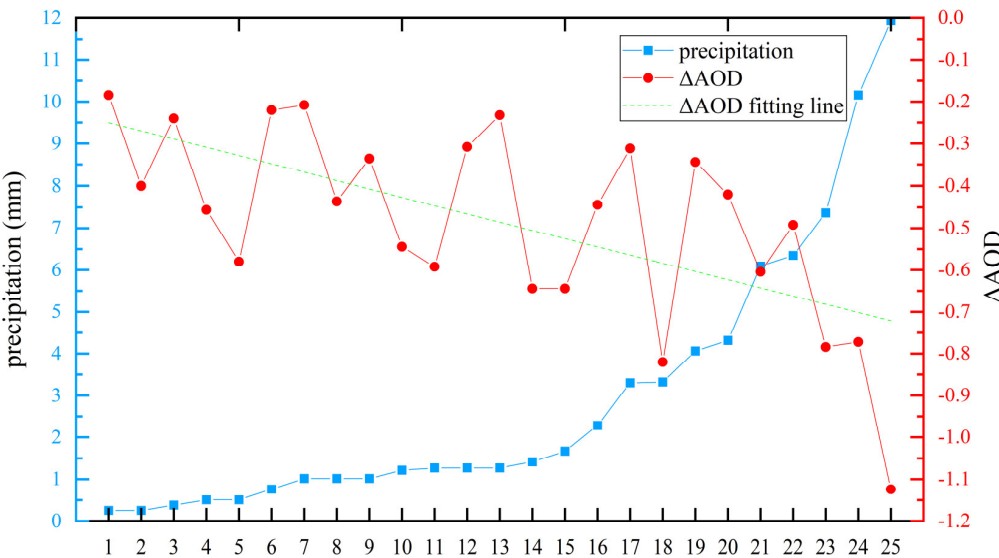

**Figure 14.** Variation relationship between precipitation and ΔAOD. In the figure, the horizontal axis presents the 25 valid data points, the left vertical axis shows the precipitation, and the right vertical axis shows the daily variation of AOD. The dashed line in the figure is the fitted line of ΔAOD.

It can be seen from Figure 14 that ΔAOD and precipitation present a negative correlation. The precipitation at points 1–15 is low, and the ΔAOD changes slowly. The precipitation at point 15–22 increased gradually, and the ΔAOD changed gradually. The ΔAOD mutation was reduced to a very low level due to heavy precipitation at points 22 to 25. This indicates that the greater the precipitation, the more obvious the negative correlation.

## 4. Conclusions

In this study, the aerosol evolution and influencing factors during typical haze periods in the Guanzhong area were analyzed. GDSA data and the HYSPLIT model were used to analyze the air trajectory of the haze period in the study area, and the change rule of air quality was analyzed combined with AQI data. The AOD of the study area was retrieved

using MODIS data and the DT algorithm and statistically analyzed based on the season. The PHA algorithm was combined with the inverted AOD, population data, GDP data, and GDEM2 data to explore the factors that influence the spatial distribution. Using AOD, precipitation, wind speed, and wind direction data from NOAA-NCEI, the influencing factors of aerosol change over time were analyzed. The results show that:

(1)　According to AQI and air mass trajectory, the haze period is divided into locally generated and externally transported. It was determined that 94.7% of the haze was locally generated haze in Guanzhong.

(2)　The AOD of the Guanzhong area was higher in winter (0.39) and spring (0.37) and lower in summer (0.20) and autumn (0.14). The spatial distribution of AOD was high in the central and eastern areas and low in the rest of the areas.

(3)　By comparing the Hamming distance, it was concluded that the aerosol spatial distribution in the study area was most strongly correlated with the following factors: GDP > population density > topography.

(4)　The significant increase in AOD was mainly caused by low wind speed, whereas the significant decrease was mainly caused by high wind speed and precipitation. When air quality improves, the wind direction is mostly from the west. The Pearson r between wind speed and AOD change was −0.63, and between precipitation and AOD change it was −0.66, both of which showed strong negative correlations.

In summary, this study used multi-source data, the HYSPLIT model, the PHA algorithm, and the DT algorithm to analyze the evolution and influencing factors of aerosols during the haze period in the Guanzhong. The results provide a reference for improving air quality and public health in the Guanzhong area.

**Author Contributions:** Conceptualization, Y.Z. and J.K.; methodology, Y.Z. and J.K.; software, Y.J. and Q.Z.; validation, Y.Z., H.M. and Q.Z.; formal analysis, Y.Z., X.W., and Y.J.; investigation, Y.Z. and Y.J.; resources, J.K.; data curation, Y.Z.; writing—original draft preparation, Y.Z.; writing—review and editing, Y.Z.; visualization, Q.Z., H.M., and X.W.; supervision, J.K.; project administration, J.K..; funding acquisition, J.K. All authors have read and agreed to the published version of the manuscript.

**Funding:** This research was funded by the Department of Science and Technology of the Shaanxi Province key research and development projects, grant number 2020ZDLSF06-07.

**Institutional Review Board Statement:** Not applicable.

**Informed Consent Statement:** Not applicable.

**Data Availability Statement:** Not applicable.

**Conflicts of Interest:** The authors declare no conflict of interest.

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
