# Peer review of "Aerosol Evolution and Influencing Factor Analysis during Haze Periods in the Guanzhong Area of China Based on Multi-Source Data"

_atmosphere, doi:10.3390/atmos13121975_

Round 1
Reviewer 2 Report
In this paper has been studied the air mass transport trajectory during this haze period (2018-2021) in Guanzhong area. Correlations between aerosols and the wind direction, wind speed, and precipitation were analyzed using weather station data. The topic is very interesting and in the complex the paper is well organized and of good quality. However, some suggestions to further improve the quality are reported below.
The introduction is quite complete, however more references can be added in the context of health application. For example the aerosol prediction can be used to prevent the diffusion of disease as the recent COVID pandemic. As regards I suggest to add:
· - Cravero C, Marsano D. Simulation of COVID-19 indoor emissions from coughing and breathing with air conditioning and mask protection effects. Indoor and Built Environment. August 2021. doi:10.1177/1420326X211039546
In that work a URANS CFD approach has been used to simulate the dispersion from the mouth of saliva droplet aerosol in closed environments with air conditioning systems. Lagrangian and Eulerian approaches have been used.
The research process has been well resumed with a diagram, such as the geographical area and its data for the following analysis. The methods used should be rearranged, by simplifying the treatment hardly readable in some parts. I suggest adding also a validation section of the approaches.
The results section is clear in all the different considered aspects, maybe I suggest enlarging some figures to be clearer. The conclusions are well resumed.
Round 2
Reviewer 2 Report
All my suggestions and questions have been answered or added in the revised work. Now the paper has increased its quality and it is ready for the publication already in this form.